# The Impact of Periodontal Inflammation on Endothelial Function Assessed by Circulating Levels of Asymmetric Dimethylarginine: A Single-Blinded Randomized Clinical Trial

**DOI:** 10.3390/jcm11144173

**Published:** 2022-07-18

**Authors:** Biagio Rapone, Elisabetta Ferrara, Erda Qorri, Gianna Dipalma, Antonio Mancini, Massimo Corsalini, Massimo Del Fabbro, Antonio Scarano, Gianluca Martino Tartaglia, Francesco Inchingolo

**Affiliations:** 1Interdisciplinary Department of Medicine, “Aldo Moro” University of Bari, 70121 Bari, Italy; giannadipalma@tiscali.it (G.D.); dr.antonio.mancini@gmail.com (A.M.); massimo.corsalini@uniba.it (M.C.); francesco.inchingolo@uniba.it (F.I.); 2Complex Operative Unit of Odontostomatology, Hospital S.S. Annunziata, 66100 Chieti, Italy; igieneeprevenzione@gmail.com; 3Dean Faculty of Medical Sciences, Albanian University, Bulevardi Zogu I, 1001 Tirana, Albania; erda79@yahoo.com; 4UOC Maxillo-Facial Surgery and Dentistry, Department of Biomedical, Surgical and Dental Sciences, School of Dentistry, Fondazione IRCCS Ca Granda, Ospedale Maggiore Policlinico, University of Milan, 20100 Milan, Italy; massimo.delfabbro@unimi.it (M.D.F.); gianluca.tartaglia@unimi.it (G.M.T.); 5Department of Oral Science, Nano and Biotechnology and CeSi-Met University of Chieti-Pescara, 66100 Chieti, Italy; ascarano@unich.it

**Keywords:** endothelial dysfunction, periodontitis, periodontal disease, periodontal therapy, oxidative stress, biomarkers

## Abstract

**Background:** Endothelial dysfunction is one of the early pathogenic events of the atherosclerotic process. Severe periodontitis is considered to be an independent contributing risk factor for the pathophysiology of endothelial dysfunction. High blood concentration of asymmetric dimethylarginine (ADMA), an L-arginine analogue that inhibits nitric oxide (NO) formation, has emerged as one of the most powerful independent risk predictors of cardiovascular disease. Abrogation of periodontal inflammation might have clinical relevance, affecting the ADMA. Insufficient clinical evidence exists for drawing clear conclusions regarding the long-term effects of periodontal disease on endothelial function, and even less evidence is available specifically on ADMA concentrations and their relationship with periodontitis. The objective of this study was to evaluate the effects of intensive periodontal treatment in modulating the endothelial function via the assessment of plasma ADMA concentration in patients diagnosed severe periodontitis. **Methods:** This was a 6-month randomized controlled trial, including 140 patients between 41 and 63 years old who were diagnosed with severe periodontitis, free from cardiovascular disease (CVD), and had traditional cardiovascular risk factors. All patients underwent a complete medical and clinical periodontal examination, a laboratory analysis of ADMA, and an ultrasound assessment of FMD of the right brachial artery. After the screening, they were randomly assigned to receive either intensive periodontal treatment (test group, *n* = 70) or community-based periodontal care (control group, *n* = 70). A full examination was carried out at baseline, 3 and 6 months after the periodontal treatment. **Results:** A total of 236 individuals diagnosed with periodontitis were screened. One hundred forty participants were enrolled. No statistically significant difference was observed over the time in ADMA concentration after the intensive periodontal treatment within the test group. No differences were revealed between the groups in the ADMA concentration at baseline and during follow-up. **Conclusions:** Intensive periodontal treatment does not affect the plasma levels of ADMA in patients without any risk for cardiovascular disease.

## 1. Introduction

Periodontitis is an infectious disease characterized by local chronic inflammation. The progressive structural damage in alveolar bone is the most significant manifestation pathway for exposure to periodontal pathogens, concomitant with the sustained inflammatory response [1,2]. Periodontitis is recognized as a promoter of chronic low-grade systemic inflammatory reaction, defined by increased levels of circulating C-reactive protein (CRP), which may support the development of cardiovascular disease, type 2 diabetes, and other relevant pathologies [3,4,5,6,7]. Severe periodontitis is considered to be an independent contributing risk factor of the pathophysiology of endothelial dysfunction [8,9,10,11]. Endothelial dysfunction is one of the early pathogenic events in the pathophysiology of atherosclerotic progression, and it has been associated with an imbalance in pro- and antioxidant activity and low chronic inflammation. There is clinical evidence to suggest an inverse association between serum asymmetric dimethylarginine (ADMA) levels and endothelial function in the general healthy population as well as in patients with coronary atherosclerosis or with multiple risk factors [12,13,14,15]. Asymmetric dimethylarginine (ADMA), a competitive inhibitor of endothelial nitric oxide synthase, is considered a likely modulation factor of endothelial dysfunction [16,17]. It results from the degradation of methylated arginine residues in proteins by protein methyltransferases, and it is metabolized by the enzyme NG-dimethylarginine dimethylaminohydrolase 1 and 2 (DDAH). During the proteolytic breakdown of protein, ADMA is released. It was independently associated with near-future cardiovascular disease (CVD) events in patients with cardiovascular risk factors, and its elevation was associated with endothelial dysfunction [18,19]. Evidence from cultured human endothelial cells suggested that the concentration-dependent superoxide regulates the ADMA levels, triggering the activation of NF-κB-mediated transcription of pro-inflammatory mediators and concomitant up-regulation of endothelial adhesions molecules and monocyte adhesion. ADMA levels are observed to be persistently increased during a variety of chronic inflammatory conditions, such as untreated essential hypertension, inflammatory bowel diseases, and glucose intolerance. It was demonstrated in vitro that ADMA was able to stimulates the release of tumor necrosis factor-α (TNF-α) in monocytes [20]. Conversely, the production of reactive oxygen species, the key signalling molecules that play an important role in initiating and sustaining the progression of inflammatory disorders, inhibits the ADMA-degrading enzyme dimethylarginine dimethylaminohydrolase (DDAH) activity, which favours the ADMA accumulation [21]. Following that, ADMA triggers the activation of the canonical NF-κB pathway, which induces the production of neutrophils with pro-inflammatory agonists, such as TNF-α and IL-8 in the innate immune system [22]. ADMA spreads the endothelial activation, characterized by a changing vasomotor activity via the release of endothelium-derived factors and the expression of cell adhesion molecules, inflammatory cytokines, and chemokines [23]. The nature of the relationship between asymmetric dimethylarginine (ADMA) and C-reactive protein (CRP) is context-dependent. Elevated blood concentrations of ADMA explain the occurrence of endothelial dysfunction through the impairment of vascular function [18]. There is evidence that periodontitis is linked with endothelial dysfunction. The biological plausibility of the association between periodontitis and endothelial dysfunction is exemplified by two potential pathways: (1) the dissemination of periodontal pathogens through blood and their invasion within endothelial cells, promoting platelet aggregation, foam-cell formation, and the development of atheroma; (2) the amplification of systemic inflammation, which directly exacerbates a pre-existent inflammatory setting [24,25,26]. Several studies have demonstrated that periodontal treatment is an effective resource for improving endothelial dysfunction [27,28,29]. However, the results of prospective studies investigating its potential beneficial effects on asymmetric dimethylarginine (ADMA) are lacking. The purpose of this trial was to investigate if cumulative inflammatory burden is associated with ADMA levels in healthy subjects without cardiovascular risk factors and if intensive periodontal treatment is effective in reducing ADMA plasma concentration.

## 2. Materials and Methods

### 2.1. Ethics

Prior to undertaking the investigation, ethical clearance was obtained from the Institutional Review Board at Albania University, Tiran, Albania (reference number 2018/16). This study was executed in accordance with the guidelines of the Declaration of Helsinki. The trial is registered with the ISRCTN registry, number ISRCTN13626790. The recruited participants provided written consent before participation.

### 2.2. Study Design

A prospective, outcome assessors-blinded, randomized controlled trial was conducted. The experimental design consisted of two groups, in which each participant received either intensive periodontal treatment or a community-based periodontal treatment in a parallel arm design, with a six-month follow-up (Figure 1).

The study was conducted from June 2018 to August 2019. The exclusion criteria included those presenting with dental implants, orthodontic bands, and ulcers; patients who received periodontal treatment within 12 months prior to the start of the study; and patients who received systemic antibiotics within the last 6 months. Pregnant or breastfeeding mothers were excluded from participating the research. Patients with a history of diabetes and those with a fasting glucose level greater than 126 mg/dl or a history of cardiovascular disease were not included. Patients with any systemic disease (e.g., diabetes mellitus or cardiovascular, kidney, liver, or lung disease) were excluded. To minimize any confounding effect of conditions that may influence endothelial function, patients with cardiovascular risk factors and active smokers were excluded. The criteria for selecting the subjects were as follows: the assessments for each patient included a detailed medical and clinical history, a full periodontal examination, blood collection, and flow-mediated brachial artery dilatation. After applying the inclusion and exclusion criteria, the participants were randomized to receive intensive periodontal treatment (IPT) or community-based periodontal treatment (CPT).

### 2.3. Sample Size Calculation

Prior to starting the trial, a power analysis for sample size calculation was employed. Based on a previous study [29], the sample size was determined by using a number of groups equal to two, an effect size of 0.30 for ADMA, a two-sided α error of 0.05, and 80% power. Assuming a 20% dropout rate, it was estimated that ≈ 70 patients per group would be needed.

### 2.4. Participants and Assessment of Periodontitis Severity

The diagnosis of periodontitis was formulated recording the following clinical periodontal variables: probing pocket depth (PPD), defined as the distance from the gingival margin to the deepest part of the probable crevice; clinical attachment level (CAL), determined by calculating the distance between the base of the pocket and the cemento–enamel junction; and bleeding on probing (BOP). Measurements were executed on six sites per tooth by employing the standard manual probe (PCPUNC15 (HuFriedy, Tuttlingen, Germany)) at a pressure of 0.25N. The severity of the periodontitis was determined according to the Consensus Report of World Workshop on the Classification of Periodontal and Peri-Implant Diseases and Conditions [30]. Periodontal measurements were carried out by dental health professionals (B.R., A.S.). The intra- and inter-rater reliability were assessed using the intra-class correlation coefficient [95% confidence interval (95% CI) 0.981–0.990]; *p* < 0.0001.

### 2.5. Laboratory Measurements

Estimations of the serum levels of ADMA were performed and analyzed in the central hospital at the onset of the study, at 3 months, and at the end of the 6-month trial for each patient. All blood samples were collected after a 12 h overnight fast. The plasma samples were thawed at room temperature for 30 min and centrifuged at a speed of 1500× *g* for 10 min, and the plasma was stored frozen in −70 °C until analysis. The serum levels of ADMA were determined using an enzyme-linked immunosorbent assay (ELISA, DLD Diagnostika, Hamburg, Germany).

### 2.6. Flow-Mediated Dilation

The measurement of endothelial function was performed using the flow-mediated dilation (FMD) technique according to current guidelines [31]. It was executed by the operator who was blinded to the randomization status of the participants, employing ultrasound imaging (Acuson XP 128/10, Siemens, Malvern, PA, USA) with the use of a 7 MHz linear probe and automated vessel diameter measurements (Brachial Tools, version 3.2.6, Medical Imaging Applications, Coralville, IA, USA) as previously described [30]. To standardize the examination conditions, the patients avoided food for 8–12 h before the procedure started. The patients were positioned with their right arm extended at an angle of ~80° from the torso and comfortably immobilized in the extended position with foam supports. A pneumatic tourniquet was inflated to 200 mmHg with the obliteration of the radial pulse. After cuff deflation, the image of the brachial artery was recorded continuously by the US from 30 s and up to 3 min. The FMD was calculated as the percentage increase in the diameter of the artery after the application of the pressure stimulation.

### 2.7. Periodontal Treatment

Before the treatment, all participants at the study received oral hygiene instructions. All patients received the IPT or CPT within 1 month from the baseline visit. Intensive periodontal treatment consisted of a single session of whole-mouth supragingival and subgingival scaling and root planing of the teeth under local analgesia within 24 h. The scaling was performed using ultrasonic instruments (EMS Piezon, Nyon, Switzerlnd) according to the manufacturer’s instructions. Gracey curettes (Hu-Friedy, Chicago, IL, USA) were used for the manual instrumentation of periodontal pockets. In addition, periodontal surgery was executed for periodontal pockets > 5 mm or residual periodontal pockets. The control group underwent CPT treatment, which consisted of a single session of supragingival scaling of the teeth [32]. The root planing was delayed at 6 months, after the completion of the trial.

### 2.8. Periodontal Disease Monitoring

All subjects underwent 3 months of periodontal monitoring and supporting periodontal therapy (SPT) for maintaining the dentition. They were instructed to maintain effective personal oral hygiene.

### 2.9. Statistical Analysis

The results for the continuous variables were expressed as mean ± standard deviation, whereas the categorical variables were given as frequencies and percentages. The normality of continuous parameters distribution was checked using the Skewness–Kurtosis test. The statistical significance was identified using analysis of variance and independent sample *t*-tests as appropriate. Shapiro–Wilk tests were conducted to determine whether variables could have been produced by a normal distribution for each category of treatment. Levene’s test was conducted to assess if the variance of variables was equal between the groups. The relationship between ADMA, FMD, PPD, and CAL was examined using the Pearson’s correlation coefficient. Cohen’s standard was used to evaluate the strength of the relationship. A mixed-model analysis of covariance (ANCOVA) with one within-subjects factor and one between-subjects factor was conducted to determine if significant differences exist between ADMA, CAL, and PPD, between the levels of treatment, after controlling for FMD. A Tukey’s post-hoc test based on an alpha of 0.05 was used to test the differences in the estimated marginal means for the effects of each combination of between-subject and within-subject. The assumption of normality was assessed by plotting the quantiles of the model residuals against the quantiles of a chi-square distribution, also called a Q–Q scatterplot. Figure 2 presents a Q–Q scatterplot of the model residuals. The homoscedasticity was evaluated by plotting the residuals against the predicted values.

The *p*-values for the within-subjects factor and the interactions with the within-subjects factor were calculated using the Greenhouse–Geisser correction to adjust for the violation of the sphericity assumption. According to Greenhouse and Geisser [33], this is the appropriate way to adjust for violations of the sphericity assumption. Figure 3 presents a scatterplot of the predicted values and model residuals.

Mauchly’s test was used to assess the assumption of sphericity. To identify influential points in the residuals, Mahalanobis distances were calculated and compared to a χ^2^ distribution.

## 3. Results

The trial recruitment began in June 2018, and the trial was concluded in August 2019. The screening phase for eligibility encompassed 236 potential participants, and 140 subjects (*n* = 70 per group) were enrolled in the trial (Figure 1, CONSORT diagram). All patients completed the study. The summary statistics can be found in Table 1.

The result of the two-tailed independent samples *t*-test was not significant based on an alpha value of 0.05, t = 0.45, and *p* = 0.235. This finding suggested that the mean of ADMA at baseline was not significantly different between the test and control groups. The mean of FMD at baseline was not significantly different between the groups based on an alpha value of 0.05, t(40) = −1.00, and *p* = 0.324. At 3 months after the treatment, the difference between the groups was not statistically significant for ADMA concentrations (t = 1.03, *p* = 0.152), while the difference between FMD values was statistically significant (t = −3.07, *p* = < 0.001). The mean of ADMA at 6 months was not significantly different between the treatment groups, with t = 1.04 and *p* = 0.306, as well as the values related to FMD (t = −0.20, *p* = 0.845). The results are presented in Table 2.

### 3.1. Mixed-Model ANCOVA Results

The results were examined based on an alpha of 0.05. The main effect for the groups was not significant (F(1, 19) = 0.31, *p* = 0.586), indicating the levels of ADMA, CAL, and PPD for the groups were all similar after controlling for FMD between the groups at 3 months. The covariate FMD at 3 months was not significantly related to ADMA, CAL, and PPD (F(1, 19) = 2.10, *p* = 0.164). The main effect for the within-subjects factor was not significant (F(2, 38) = 0.95, *p* = 0.358), indicating the values of ADMA, CAL, and PPD were similar between the groups after controlling for FMD at 6 months. The interaction effect between the within-subjects factor and the group was not significant (F(2, 38) = 0.02, *p* = 0.930), indicating that the relationships between ADMA, CAL, and PPD were similar between the groups after controlling for FMD at 3 and 6 months. The interaction effect between the within-subjects factor and FMD_3_months was not significant (F(2, 38) = 1.83, *p* = 0.189), indicating that the relationships between ADMA_3_months, CAL_3_months, and PPD_3_months were similar for all values of FMD_3_months. The interaction effect between the within-subjects factor and the group was not significant (F(2, 40) = 0.11, *p* = 0.790), indicating that the relationships between ADMA, PPD, and CAL were similar between the groups after controlling for FMD at 6 months. The main effect for the within-subjects factor was not significant (F(2, 88) = 0.87, *p* = 0.421), indicating the values of ADMA at each timepoint were all similar. The interaction effect between the within-subjects factor and the group was significant (F(2, 88) = 5.63, *p* = 0.005), indicating that the relationships between ADMA differed significantly between the groups over time. Table 3 presents the ANCOVA results.

The mean contrasts utilized Tukey comparisons based on an alpha of 0.05. Tukey comparisons were used to test the differences in the estimated marginal means for each combination of between-subject and within-subject effects.

Regarding the combination of between-subjects, for the control group, the ADMA at 3 months resulted significantly less than the ADMA at 6 months (t(44) = −2.49, *p* = 0.043). For the test group, the ADMA at 6 months showed a statistically significant reduction greater than the baseline (t(44) = 2.71, *p* = 0.025). No other significant differences were found. Regarding the FMD value, the main effect for the groups was not significant (F(1, 40) = 0.23, *p* = 0.634), indicating that the levels of the groups were all similar for FMD at baseline and over time. The main effect for the within-subjects factor was significant (F(2, 80) = 27.92, *p* < 0.001), indicating there were significant differences between the values of FMD at baseline and at 3 and 6 months after the treatment. The interaction effect between the within-subjects factor and the group was not significant (F(2, 80) = 1.56, *p* = 0.221), indicating that the relationships between the FMD percentage at each timepoint were similar between the levels of the groups. The test of the between-subjects effects for the control group showed that the FMD at baseline was significantly less than the FMD at 6 months (t(40) = −4.51, *p* < 0.001). For the test group, the FMD at baseline was significantly less than the FMD at 3 months (t(40) = −4.07, *p* < 0.001) as well as at 6 months (t(40) = −7.11, *p* < 0.001). No other significant differences were found between the groups. Table 4 presents the marginal means contrasts for the mixed-model ANCOVA.

### 3.2. Pearson’s Correlation Analysis Results

The result of the correlations was examined using Holm’s correction to adjust for multiple comparisons based on an alpha value of 0.05. We observed a significant positive correlation in the test group between CAL and PPD at baseline, with a correlation of 0.88, indicating a large effect size (*p* < 0.001, 95.00% CI = [0.73, 0.95]). No other significant correlations were found. No significant correlations were observed between each pair of variables for the control group. The results of the correlations are shown in Table 5.

As shown in Table 6, no significant correlations were revealed at 3 months between any pairs of variables for each group.

At 6 months, no statistically significant correlations between any pairs of variables were revealed, as shown in Table 7.

## 4. Discussion

Periodontitis, along with the local immune response, can lead to an increased systemic inflammatory burden and activate the systemic inflammatory response [7]. It is generally accepted that periodontitis is dimensional and that its severity can impact systemic pathological conditions having a core common theme: inflammation [34,35]. A growing body of literature has examined the link between endothelial dysfunction and periodontitis, and several studies reported findings that were consistent with an impairment of endothelial function in subjects affected by moderate or severe periodontitis [36,37]. Prior research has found that periodontitis is associated with worsening endothelial function over time in patients with pre-existent cardiovascular disease or cardiovascular risk [37]. Employing laboratory measures, several researchers demonstrated that chronic periodontal infection leads to worsen function of the vascular endothelium. Tonetti et al. [36] examined the flow-mediated dilatation, inflammatory biomarkers and markers of coagulation, and endothelial activation. The findings revealed that intensive periodontal treatment improved short-term endothelial function when compared with conventional treatment. These findings are consistent with data reported by Seinost et al. [37], who showed that flow-mediated dilation was significantly lower in patients with periodontitis when compared with healthy control (6.1% ± 4.4% vs. 8.5% ± 3.4%, *p* = 0.002), and that periodontal treatment caused a significant improvement in FMD (9.8% ± 5.7%; *p* = 0.003 in comparison to baseline values) and a simultaneously significant decrease in C-reactive protein concentrations (*p* = 0.026). To date, limited studies have examined ADMA concentrations in relation to periodontitis and endothelial function [29]. Recently, ADMA was proposed to be useful for CV risk stratification and for predicting endothelial function. The data suggest a direct action of ADMA through the increase of endothelial permeability, NF-κB signaling alteration, actin fiber reorganization, and acceleration of EC senescence [8,21]. Furthermore, it was demonstrated that this molecule promotes inflammation and endothelial dysfunction [21]. We hypothesized that chronic periodontal inflammation may accelerate endothelial dysfunction by the up-regulation of ADMA. This study then set out to assess the impact of intensive periodontal treatment in endothelial function in a setting of systemic health. Our study showed that plasma concentrations of ADMA were not elevated in patients with periodontitis. However, the intensity of treatment did not lead any significant difference in ADMA levels in the test group compared with the control group. Contrary to expectations, this study did not find a significant difference between the groups, suggesting that cardiovascular risk factors serve as a seed for the development of disfunction. Tsioufis et al. [38] conducted a trial involving untreated hypertensive subjects with concomitant periodontal disease and demonstrated the reduction of asymmetric dimethylarginine (ADMA) concentration after periodontal treatment. Okada et al. [29] recently investigated the impact of periodontal treatment on ADMA levels and FMD values. In contrast with our observations, the authors reported no significant between-group differences in FMD (mean difference, −0.2%; 95% CI, −1.4–0.9; *p* = 0.708) and serum ADMA levels (mean difference, 0.01 nmol/L; 95% CI, −0.00–0.03; *p* = 0.122). The difference with our observations may be because the participants that were selected had been diagnosed with early-stage periodontal disease. In our study, the ADMA levels of the patients in the test group were found not to be markedly higher than those in the control group at each timepoint. Our study showed that patients who received intensive periodontal treatment showed reduced plasma concentrations of ADMA and increased FMD values after 3 and 6 months of treatment, compared with the control patients. However, periodontitis at baseline did not lead to any significant difference in ADMA concentrations and FMD levels, compared with the control group, and the differences over time did not show any significant modifications in ADMA concentration. The most obvious finding to emerge from the analysis was that the periodontal parameters decreased sensibility after intensive periodontal treatment; the most clinically relevant funding was the increase of FMD for the test group after therapy. These results highlight the impact of treating periodontitis on endothelial function. Limitations exist in the current study, including the limited sample size that made it difficult to generalize results, the lack of previous research studies on the topic, and the relatively short duration of follow-up. Experimental and longitudinal studies are necessary to check the influence of periodontitis on ADMA concentrations.

## 5. Conclusions

There is scant evidence about the effects of periodontal treatment on endothelial function by means of ADMA. Despite these promising results, questions remain. Further research should be undertaken to investigate the impact of periodontal treatment on endothelial function.

## Figures and Tables

**Figure 1 jcm-11-04173-f001:**
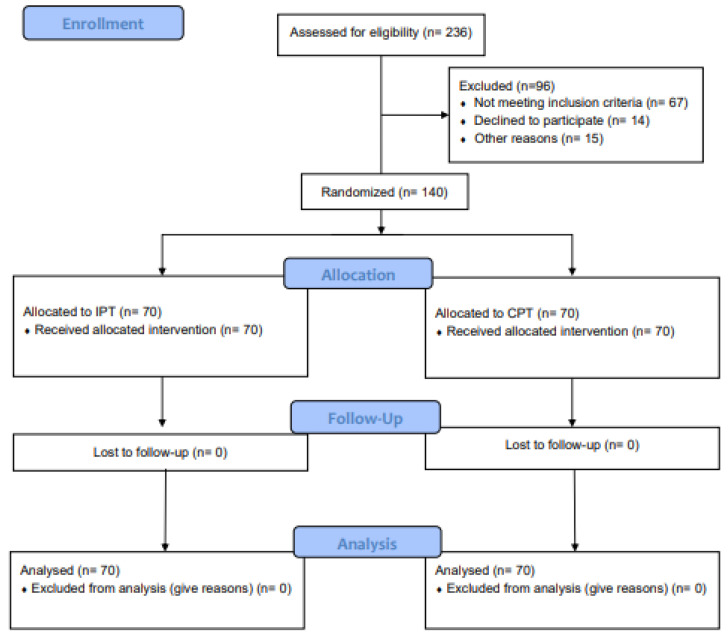
The study flowchart (flowchart randomization; CONSORT diagram). IPT: Intensive periodontal treatment; CPT: Community based periodontal treatment.

**Figure 2 jcm-11-04173-f002:**
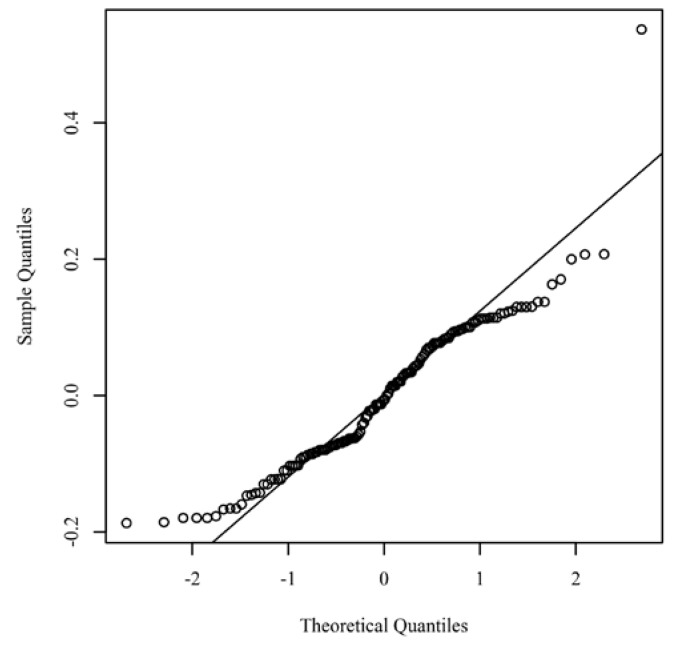
Q–Q scatterplot for normality of the residuals for the regression model.

**Figure 3 jcm-11-04173-f003:**
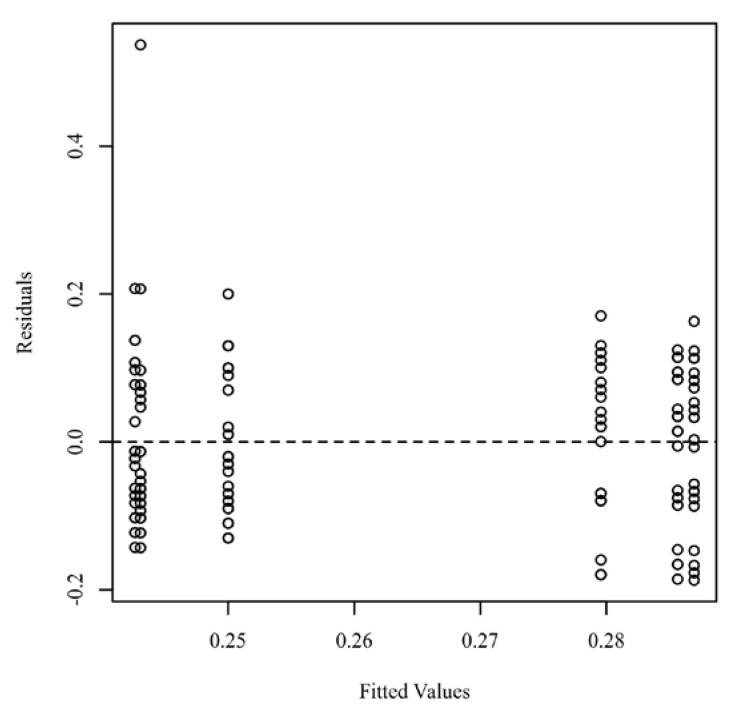
Scatterplot of the residuals testing homoscedasticity.

**Table 1 jcm-11-04173-t001:** Summary statistics table for interval and ratio variables at baseline.

Baseline		Group IPT			Group CPT		
Variable	M	SD	SE_M_	M	SD	SE_M_	*p* Value
ADMA (μmol/L)	0.413	0.06	0.01	0.393	0.05	0.01	0.235
PPD (mm)	5.73	0.50	0.10	5.64	0.39	0.08	0.831
CAL (mm)	5.87	0.48	0.10	5.9	0.51	0.11	0.615
FMD	9.81	5.23	0.04	10.11	4.32	0.07	<0.5

IPT: intensive periodontal treatment; CPT: community-based periodontal treatment; ADMA: asymmetric dimethylarginine; PPD: probing pocket depth; CAL: clinical attachment level; FMD: flow-mediated dilatation; M: mean; SD: standard deviation; SE_M_: standard error mean.

**Table 2 jcm-11-04173-t002:** Two-tailed independent samples *t*-test for ADMA by group.

		Group IPT		Group CPT				
		M	SD	M	SD	t	*p*	d
ADMA (μmol/L)	Baseline	0.413	0.06	0.393	0.05	0.45	0.235	0.36
	3 months	0.40	0.005	0.390	0.06	1.03	0.152	0.43
	6 months	0.38	0.05	0.387	0.04	1.04	0.306	0.31
FMD (%)	Baseline	9.62	0.62	9.81	0.62	−1.00	0.324	0.32
	3 months	10.01	0.59	9.82	0.56	−3.07	<0.001	0.03
	6 months	10.00	0.56	9.81	0.53	−0.20	0.845	0.06

IPT: intensive periodontal treatment; CPT: community-based periodontal treatment; ADMA: asymmetric dimethylarginine; FMD: flow-mediated dilatation; M: mean; SD: standard deviation.

**Table 3 jcm-11-04173-t003:** Mixed-model ANCOVA results.

Source	*df*	*SS*	*MS*	*F*	*p*	η_p_^2^
Group	1	0.06	0.06	0.31	0.586	0.02
FMD 3 months	1	0.40	0.40	2.10	0.164	0.10
FMD 6 months	1	0.53	0.52	2.80	0.111	0.13
Residuals	19	3.58	0.19			
**Within-Subjects**						
Within factor	2	0.45	0.22	0.95	0.358	0.05
Group: within factor	2	0.008	0.004	0.02	0.930	0.0009
FMD 3 months: within factor	2	0.87	0.43	1.83	0.189	0.09
FMD 6 months: within factor	2	0.89	0.44	1.87	0.184	0.09
Residuals	38	8.98	0.24			

df: degrees of freedom; SS: sum of squares; MS: mean square; F: f-ratio; η_p_^2^: partial Eta-squared.

**Table 4 jcm-11-04173-t004:** Marginal means contrasts for the mixed-model ANCOVA.

Contrast	Difference	SE	df	t	*p*
**Group|IPT**					
ADMA baseline–3 months	0.007	0.01	44	0.69	0.770
ADMA baseline–6 months	−0.01	0.010	44	−1.35	0.376
ADMA 3 months–6 months	−0.02	0.008	44	−2.49	0.043
FMD baseline–3 months	−0.21	0.10	40	−2.10	0.103
FMD baseline–6 months	0.32	0.07	40	−4.51	<0.001
FMD 3 months–6 months	−0.11	0.07	40	−1.65	0.238
**Group|CPT**					
ADMA baseline–3 months	0.010	0.01	44	0.88	0.655
ADMA baseline–6 months	0.03	0.01	44	2.71	0.025
ADMA 3 months–6 months	0.02	0.009	44	2.13	0.095
FMD baseline–3 months	−0.38	0.09	40	−4.07	<0.001
FMD baseline–6 months	−0.47	0.07	40	−7.11	<0.001
FMD 3 months–6 months	−0.09	0.06	40	−1.47	0.314

*Note.* Tukey comparisons were used to test the differences in the estimated marginal means. IPT: intensive periodontal treatment; CPT: community-based periodontal treatment; ADMA: asymmetric dimethylarginine; FMD: flow-mediated dilatation; SE: standard error; df: degrees of freedom.

**Table 5 jcm-11-04173-t005:** Pearson’s correlation results among PPD, CAL, ADMA, and FMD at baseline.

Combination	*r*	95.00% CI	*p*
**Group|IPT**			
CAL–ADMA	−0.08	[−0.48, 0.34]	1.000
PPD–ADMA	−0.13	[−0.52, 0.30]	1.000
CAL–FMD	0.04	[−0.39, 0.46]	1.000
PPD–FMD	0.13	[−0.31, 0.52]	1.000
CAL–PPD	0.88	[0.73, 0.95]	<0.001
**Group|CPT**			
CAL–ADMA	0.029	[−0.14, 0.63]	1.000
PPD–ADMA	0.32	[−0.10, 0.65]	1.000
CAL–FMD	0.25	[−0.22, 0.62]	1.000
PPD–FMD	−0.38	[−0.55, 0.32]	0.670
CAL–PPD	0.29	[−0.18, 0.65]	0.658

*Note*: *p*-values adjusted using Holm’s correction. IPT: intensive periodontal treatment; CPT: community-based periodontal treatment; ADMA: asymmetric dimethylarginine; FMD: flow-mediated dilatation; PPD: probing pocket depth; CAL: clinical attachment level.

**Table 6 jcm-11-04173-t006:** Pearson’s correlation results among PPD, CAL, ADMA, and FMD at 3 months.

Combination	*r*	95.00% CI	*p*
**Group|IPT**			
CAL–ADMA	0.10	[−0.32, 0.50]	0.635
PPD–ADMA	−0.23	[−0.59, 0.20]	0.587
CAL–FMD	0.18	[−0.28, 0.58]	1.000
PPD–FMD	−0.09	[−0.51, 0.37]	1.000
CAL–PPD	0.28	[−0.15, 0.62]	0.590
**Group|CPT**			
CAL–ADMA	−0.34	[−0.66, 0.08]	1.000
PPD–ADMA	0.16	[−0.54, 0.27]	1.000
CAL–FMD	0.18	[−0.28, 0.58]	1.000
PPD–FMD	−0.12	[−0.53, 0.34]	1.000
CAL–PPD	0.29	[−0.14, 0.63]	1.000

*Note*: *p*-values adjusted using Holm’s correction. IPT: intensive periodontal treatment; CPT: community-based periodontal treatment; ADMA: asymmetric dimethylarginine; FMD: flow-mediated dilatation; PPD: probing pocket depth; CAL: clinical attachment level.

**Table 7 jcm-11-04173-t007:** Pearson’s correlation results among PPD, CAL, ADMA, and FMD at 6 months.

Combination	*r*	95.00% CI	*p*
**Group|IPT**			
CAL–ADMA	0.10	[−0.49, 0.32]	0.643
PPD–ADMA	−0.22	[−0.58, 0.21]	0.606
CAL–FMD	0.13	[−031, 0.52]	1.000
PPD–FMD	0.03	[−0.40, 0.44]	1.000
CAL–PPD	0.30	[−0.12, 0.64]	0.471
**Group|CPT**			
CAL–ADMA	0.23	[−0.20, 0.59]	1.000
PPD–ADMA	−0.18	[−0.55, 0.25]	1.000
CAL–FMD	0.03	[−0.42, 0.46]	1.000
PPD–FMD	−0.09	[−0.51, 0.36]	1.000
CAL–PPD	0.20	[−0.23, 0.57]	1.000

*Note*: *p*-values adjusted using Holm’s correction. IPT: intensive periodontal treatment; CPT: community-based periodontal treatment; ADMA: asymmetric dimethylarginine; FMD: flow-mediated dilatation; PPD: probing pocket depth; CAL: clinical attachment level.

## Data Availability

Not applicable.

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
