# Peer review of "The Impact of Periodontal Inflammation on Endothelial Function Assessed by Circulating Levels of Asymmetric Dimethylarginine: A Single-Blinded Randomized Clinical Trial"

_jcm, 2022, doi:10.3390/jcm11144173_

Round 1

Reviewer 1 Report

This study is a meaningful study to evaluate the effectiveness of periodontitis diagnosis through ADMA and FDA evaluation. However, it seems necessary to consider statistical analysis to improve the readability of the research results. Specific explanations are as follows:

1. Please describe the group name consistently as IPT and CPT in all tables as shown in Figure 1.

 2. The main dependent variable of this study is to evaluate plasma ADMA concentration. In Table 1, which is the result of the homogeneity test between groups, there is a significant difference between groups in FMD, so an analysis of the difference in ADMA after intervention (3, 6 months) requires an analysis that controlled the FDA. In this case, ANCOVA analysis seems appropriate. In the case of Table 3, if ADMA and FDA are analyzed separately, but FDA is controlled and the results of analysis of the effects of ADMA are presented, it seems that readability will increase. We recommend that you seek advice from a statistician.

 3. Table 5-9 Correlation analysis results, IPT and CPT results are revised into one table and presented in each table at baseline, 3 months, and 6 months later, so that readability is improved.

Author Response

# Reviewer 1

This study is a meaningful study to evaluate the effectiveness of periodontitis diagnosis through ADMA and FDA evaluation. However, it seems necessary to consider statistical analysis to improve the readability of the research results. Specific explanations are as follows:

  1. Please describe the group name consistently as IPT and CPT in all tables as shown in Figure 1.

-Thank you so much for your input. The group name has been added in the manuscript.

  1. The main dependent variable of this study is to evaluate plasma ADMA concentration. In Table 1, which is the result of the homogeneity test between groups, there is a significant difference between groups in FMD, so an analysis of the difference in ADMA after intervention (3, 6 months) requires an analysis that controlled the FDA. In this case, ANCOVA analysis seems appropriate. In the case of Table 3, if ADMA and FDA are analyzed separately, but FDA is controlled and the results of analysis of the effects of ADMA are presented, it seems that readability will increase. We recommend that you seek advice from a statistician.

-The authors would like to thank the reviewer for the very supportive comments. Our statistician has conducted the ANCOVA, controlling for FMD. Please, find the manuscript corrected.  

3.1. Mixed Model ANCOVA Results

The results were examined based on an alpha of 0.05. The main effect for Group was not significant, F(1, 19) = 0.31, p = .586, indicating the levels of Group were all similar for ADMA, CAL, and PPD after controlling for FMD between the levels of Group, at 3 months. The covariate, FMD at 3 months, was not significantly related to ADMA, CAL, and PPD, F(1, 19) = 2.10, p = .164. The main effect for the within-subjects factor was not significant, F(2, 38) = 0.95, p = .358, indicating the values of ADMA, CAL, and PPD were similar between the levels of Group after controlling for FMD at 6 months. The interaction effect between the within-subjects factor and Group was not significant, F(2, 38) = 0.02, p = .930, indicating that the relationships between ADMA, CAL, and PPD were similar between the levels of Group after controlling for FMD at 3 and 6 months. The interaction effect between the within-subjects factor and FMD_3_months was not significant, F(2, 38) = 1.83, p = .189, indicating that the relationships between ADMA_3_months, CAL_3_months, and PPD_3_months were similar for all values of FMD_3_months.The interaction effect between the within-subjects factor and Group was not significant, F(2, 40) = 0.11, p = .790, indicating that the relationships between ADMA, PPD, and CAL were similar between the levels of Group after controlling for FMD at 6 months. The main effect for the within-subjects factor was not significant, F(2, 88) = 0.87, p = 0.421, indicating the values of ADMA at each time-point were all similar. The interaction effect between the within-subjects factor and Group was significant, F(2, 88) = 5.63, p = 0.005, indicating that the relationships between ADMA differed significantly between the levels of Group over the time. Table 3 presents the ANCOVA results.

  1. Table 5-9 Correlation analysis results, IPT and CPT results are revised into one table and presented in each table at baseline, 3 months, and 6 months later, so that readability is improved.

The authors would like to thank the reviewer for the very supportive comments.

Reviewer 2 Report

“In this manuscript, Rapone et al. demonstrate no differences in the concentrations of ADMA in the plasma of the periodontal group after 3 months and 6 months of periodontal treatment compared to baseline”

This report is interesting and potentially quite useful. Just a few comments for improvement:

1.     Introduction:

a.      Define the full name of ADMA the 1st time you name it in the introduction, line 60.

b.     Explain where ADMA comes from.

2.     Materials and Methods

a.      Explain: What intensive periodontal treatment or a community-based periodontal treatment consists of and their differences

b.     Write the ethical components of the study at the beginning of the material and method

c.      Who performed the periodontal examination, how many people performed it, and was there calibration?

d.     What was the procedure for taking the blood and who performed it?

e.     What are the detection limits of ELISA and how was the procedure performed?

3.     Results:

Aa.      Explain each abbreviation in the table below it. It is difficult to follow the results with abbreviations without explanations in the tables.

4.     Discussion: Refer to the limitations of the study

Author Response

#Reviewer 2

“In this manuscript, Rapone et al. demonstrate no differences in the concentrations of ADMA in the plasma of the periodontal group after 3 months and 6 months of periodontal treatment compared to baseline”

This report is interesting and potentially quite useful. Just a few comments for improvement:

  1. Introduction:
  2. Define the full name of ADMA the 1st time you name it in the introduction, line 60.

-Thank you for your input. We have defined the full name of ADMA. Please find the revised name in red into the text.

  1. Explain where ADMA comes from.

-Thank you for your suggestions. We have added the information, as follows: “It results from the degradation of methylated arginine residues in proteins by protein methyltransferases and it is metabolized by the enzyme NG, NG-dimethylarginine dimethylaminohydrolase 1 and 2 (DDAH). During proteolytic breakdown of protein, ADMA is released. It has been independently associated with near future cardiovascular disease (CVD) events in patients with cardiovascular risk factors, and its elevation is associated with endothelial dysfunction [18,19]. Evidence from cultured human endothelial cells suggested that superoxide concentration-dependent regulates the ADMA levels, triggering activation of NF- κB-mediated transcription of pro-inflammatory mediators and concomitant up-regulation of endothelial adhesions molecules and monocyte adhesion. 

  1. Materials and Methods
  2. Explain: What intensive periodontal treatment or a community-based periodontal treatment consists of and their differences

- Thank you for your input. The differences have been added, as follows:

-Before treatment, all participants at the study received the oral hygiene instructions. All patients received the IPT or CPT, within 1 month from the baseline visit. Intensive periodontal treatment consisted of a single session of whole mouth supragingival and subgingival scaling and root planing of the teeth under local analgesia within 24 hours. Scaling was performed by ultrasonic instruments (EMS SA, Germany) according to the manufacturer’s instructions. Gracey curettes (HU-Friedy, USA) were used for manual instrumentation of periodontal pockets. In addition, periodontal surgery was executed for periodontal pockets >5 mm or residual periodontal pockets. Control group underwent to CPT treatment, who consisted of a single session of supragingival scaling of the teeth [32]. The root planing was delayed at 6 months, after completion of the trial. The same two clinicians (B.R; A.S.) performed the treatment. Intra-examiner and inter-examiner reliability and alignment in obtaining values and positions of measurements tools were assessed.

  1. Write the ethical components of the study at the beginning of the material and method

- Thank you for your remarks. We have corrected the manuscript. We have also specified the number of trial registration, as follows:

2. Materials and Methods

2.1. Ethics

Prior to undertaking the investigation, ethical clearance was obtained from the Institutional Review Board at the Albania University, Tiran, Albania (Reference Number 2018/16). This study was executed in accordance with the guidelines of the Declaration of Helsinki. The trial is registered with the ISRCTN registry, number ISRCTN13626790 Participants recruited provided written consent before participation.

  1. Who performed the periodontal examination, how many people performed it, and was there calibration?

- Thank you for the input. We have add the information, as follows: Periodontal measurements were carried out by dental health professionals (B.R., A.S.). The intra and inter-rater reliability was assessed using the intraclass correlation coefficient [95% confidence interval (95% CI) 0.981–0.990]; p < 0.0001.

  1. What was the procedure for taking the blood and who performed it?

- Estimations of serum levels of the ADMA were performed and analyzed in the central hospital, at onset of the study, at 3 months, and at the end of the 6-month trial for each patient. All blood samples were collected after a 12 hour overnight fast. Plasma samples were thawed at room temperature for 30 min, then centrifuged at speed 1500 x g for 10 min and plasma stored frozen in -70° c until analysis. Serum levels of ADMA were determined by enzyme-linked immunosorbent assay (ELISA, DLD Diagnostika, Hamburg, Germany).

  1. What are the detection limits of ELISA and how was the procedure performed?

Thank you for your question. However, the ADMA analysis was performed and analyzed in the central hospital, and we investigated the referred data. In our opinion, it is not necessary explain the limits of ELISA. However, if the reviewer thinks it necessary, we will present the limits.

  1. Results:

Aa.      Explain each abbreviation in the table below it. It is difficult to follow the results with abbreviations without explanations in the tables.

  • Thank you for your suggestions. The manuscript has been corrected.

  1. Discussion: Refer to the limitations of the study

- Thank you for your suggestions. Please, find the corrections:

Limitations exist in the current study, beginning the limited sample size which makes it difficult to generalize results, lack of previous research studies on the topic, and the relatively short duration of follow-up. Experimental and longitudinal studies would be necessary to check the influence of periodontitis on ADMA concentration.

Round 2

Reviewer 1 Report

Overall, it was confirmed that the first review comments was reflected.

1. In the titles of tables and figures, write only proper nouns and the first letter of the sentence in uppercase, and revise the rest in lowercase.

2. Please move the title of the figure below the figure.

Best regards,

Author Response

Thank you so much for your positive feedback and your suggestions. The manuscript has been corrected.